# Study of a New Wave Energy Converter with Perturb and Observe Maximum Power Point Tracking Method

José Carlos Ugaz Peña , Christian Luis Medina Rodríguez and Gustavo O. Guarniz Avalos *

Faculty of Engineering, Universidad Tecnológica del Perú, Lima 15046, Peru; josecarlos84@gmail.com (J.C.U.P.); clmr.urt@gmail.com (C.L.M.R.)
* Correspondence: gguarniz@utp.edu.pe

**Abstract:** Ocean waves contain the highest energy density among renewable energy sources. However, harnessing the energy from ocean waves represents a challenge because wave energy converters (WECs) must be designed to have great survivability and efficiency. The power production challenge of any WEC depends on the power take-off (PTO) system efficiency. Maximum power point tracking (MPPT) algorithms have been widely applied in renewable energy from photovoltaic and wind sources, and have subsequently been adapted to wave energy converters (WECs). Energy extraction is optimized by applying MPPT, resulting in an increase in efficiency. This study aims to address the analysis of the influence of the perturb and observe MPPT in the electrical power performance of a WEC composed of a point absorber, a hinged arm and a direct mechanical drive PTO system. The PTO is characterized by a pulley system, a counterweight, one-way bearings, a gearbox, a flywheel and an electric generator; in the present study it is considered to be a cylindrical point absorber. The linear theory and the viscous damping effect are applied to analyze the hydrodynamic behavior of the point absorber. Regarding the two generators considered in the present study, the contribution of MPPT is greater for the low power generator; the high values of the capture width ratio (CWR) occur at low values of period and wave height, showing the maximum value in the high-power generator.

**Keywords:** wave energy converter; direct mechanical drive power take-off; gearbox; pulley system; maximum power point tracking

## 1. Introduction

The use of fossil fuels is the main cause of environmental damage and renewable energies represent the main and safest alternative to counteract the effects of fossil fuels [1,2]. The most available renewable energy is marine energy; the ocean makes up 71% of the Earth and it could meet the electricity demand of the whole world [3]. The motion of gravity waves is studied through different numerical methods [4–6]. Wind and solar energy have undergone great technological development; however, the production of wave energy is more constant and predictable [7]. Waves have the highest energy density of all renewable energy sources [8]. The technological development of wave energy converters (WECs) is increasing worldwide [9]. Due to its small size, low complexity and low cost, the point absorber is the most studied WEC type [10–13]. The accessibility of these devices is important for maintenance and repair, as environmental conditions can make this task difficult [14]. For this reason, WECs based on a point absorber connected to shore via an articulated arm are generating much interest due to their easy installation and maintenance [15]. A very important component of the WEC is the power take-off (PTO) system, which represents the mechanism that transforms the kinetic energy of the component that interacts with the wave to generate electricity. The efficiency of the WEC depends mainly on this system; it can be considered as the brain [11] because it can influence the dynamics of the component that interacts with the wave to maximize energy capture. The PTO system is based on a hydraulic [16], pneumatic [17,18] or mechanical system [9,19]; hydraulic and

pneumatic components decrease the efficiency of WECs [9]. For this reason, in recent years, the application of the direct drive mechanical system and the direct electrical drive system has increased [11]. As reported in [9], PTO systems based on direct mechanical drive are very well-known and used by 31 developers from all over the world. This system can be composed of a rack and pinion, unidirectional bearings, a belt drive system, a pulley system or a screw mechanism [20–25]. The maintenance cost of these components is the challenge of this system because they are exposed to high load cycles; the size of the gearbox represents another problem in some cases [9].

The control system in WECs is introduced to maximize energy capture and provide a physical device constraint that must not be violated. This objective is achieved by the force/torque manipulation of the PTO system. However, the goal should not be maximizing conversion efficiency, but minimizing the cost of converted energy [26]. During the last few years the control system has progressed; however, there is still the challenge of reproducing the non-linearity of the WEC system with respect to the hydrodynamics and the PTO system [26,27]. One of the most important characteristics to take into account when extracting wave energy is its high variation over time. For the efficient operation of WEC systems, all the available wave energy should be converted into electricity, even in the face of a variable wave regime [28]. This variability of the available power is common to other renewable energy sources, such as photovoltaics and wind, where researchers have introduced the concept of maximum power point tracking (MPPT) to adapt the converter to the variation of the energy source and achieve higher efficiency [29,30]. Most of these methods, such as perturb and observe, are heuristics algorithms where no model of the plant is required [31,32] and optimization is based on a gradient-ascent approach [28]. MPPT methods have been adapted for WECs in [28,32–34]. The implementation of MPPT methods imposes certain behavior on the generator currents that are related to the torque and force of the PTO, thus influencing the converter performance. Furthermore, operational limits, such as speed, torque and power, are critical considerations when designing and operating electrical generators [35,36]. These limits determine the maximum output capacity and performance of a generator, and exceeding them may result in system failures, damage to equipment and even safety hazards. Speed limits, for instance, are essential to prevent mechanical stress, overheating and even rotor or stator damage. Torque limits, on the other hand, determine the maximum force that a generator can produce and transmit without causing damage or malfunction. Therefore, WECs should be designed within this operational limit to ensure reliable and safe energy production and avoid costly and dangerous accidents. Furthermore, the proposed MPPT method also considers these limits by disabling the optimization when any of them is reached.

The aim of this study is to demonstrate the new WEC's performance and the influence of the MPPT perturb and observe (P&O) method on electric power generation. The WEC is based on a cylindrical point damper, articulated arm and direct mechanical drive PTO system. The power take-off is made up of a pulley system, a counterweight, one-way bearings, a flywheel, a gearbox and an electric generator. The main components of the power take-off that influence the generation of energy, the variation of the transmission ratio and the power of the electric generator are analyzed. In the present study, regular waves of three different periods $T_p$ and heights $H_w$ are considered; linear wave theory and the effect of viscous damping are applied to calculate the hydrodynamic force at the point of absorption. This force is considered in the wave-to-wire model to describe the operation of the WEC; this explains the coupling and decoupling between the point absorber and the electric generator. The increase in power generation due to MPPT and the WEC capture width ratio are detailed.

## 2. Wave-to-Wire Model

The proposed wave energy converter is shown in Figure 1, the PTO system is composed of a pulley system, a counterweight, a gearbox, a flywheel and an electric generator, see Figure 2a. The aim of the counterweight is to maintain tension on the pulley system.

The pulley system transforms the oscillation movement of the point absorber into a rotation movement; the cable wraps around the main pulleys in order to transmit torque to the primary shaft, see Figure 2a; the main pulleys use one-way bearings to transmit rotation in one direction. The gearbox increases the rotation velocity; the output shaft of the gearbox is connected to the secondary shaft through a coupling based on one-way bearing that defines the coupling and uncoupling of the point absorber and the electric generator. An MPPT control is implemented to compute the torque of the electric generator. The flywheel is mounted on the secondary shaft to store the captured kinetic energy. The stored energy is used when the point absorber and the electric generator are uncoupled. The point absorber considered in the present work has been studied by [20] and is described in Table 1.

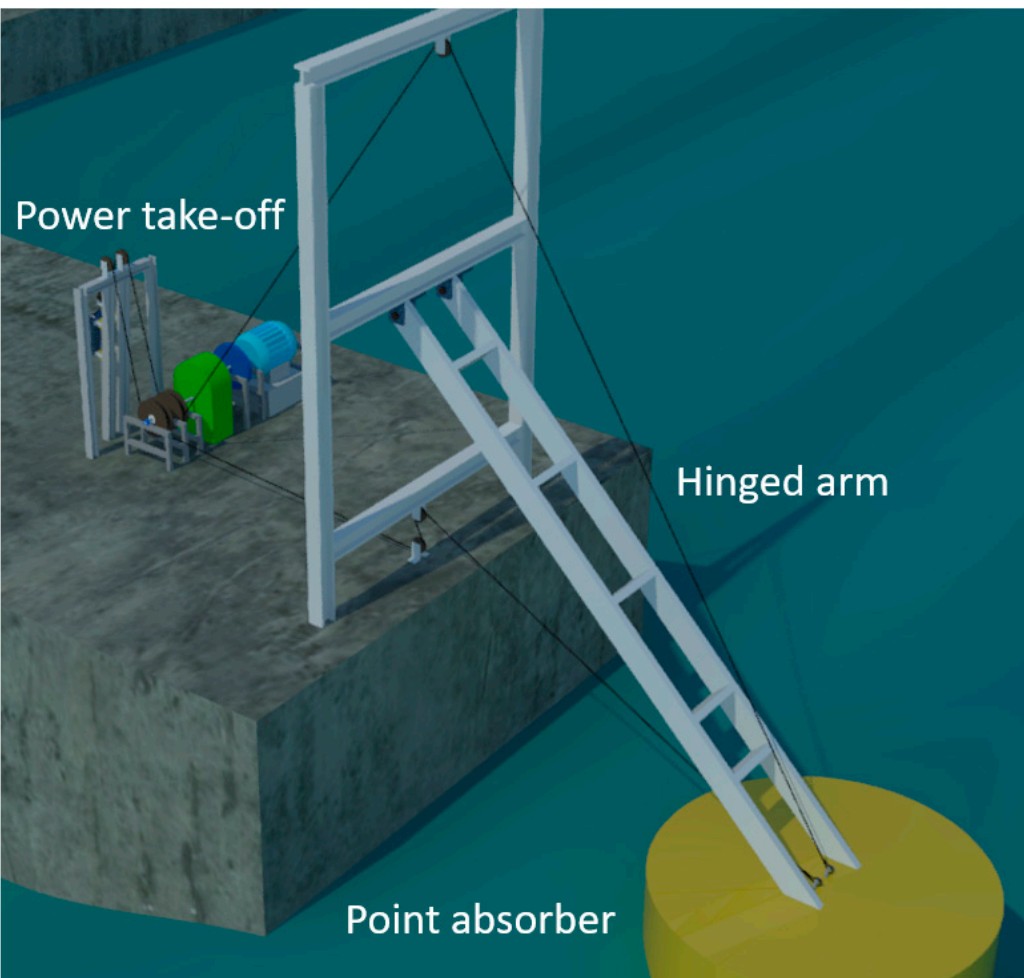

**Figure 1.** Components of the wave energy converter.

**Table 1.** Characteristics of the buoy.

| Parameter | Value |
| --- | --- |
| Mass (m) | 40.25 t |
| Diameter (D) | 5 m |
| Draft (T) | 2 m |
| Natural Period (Tn) | 3.64 s |

The complexity of the WEC operation is simplified in Figure 3. The schematic model considers a counterweight for each mean pulley, unlike Figure 2. The main dimensions of the schematic model are shown in Table 2.

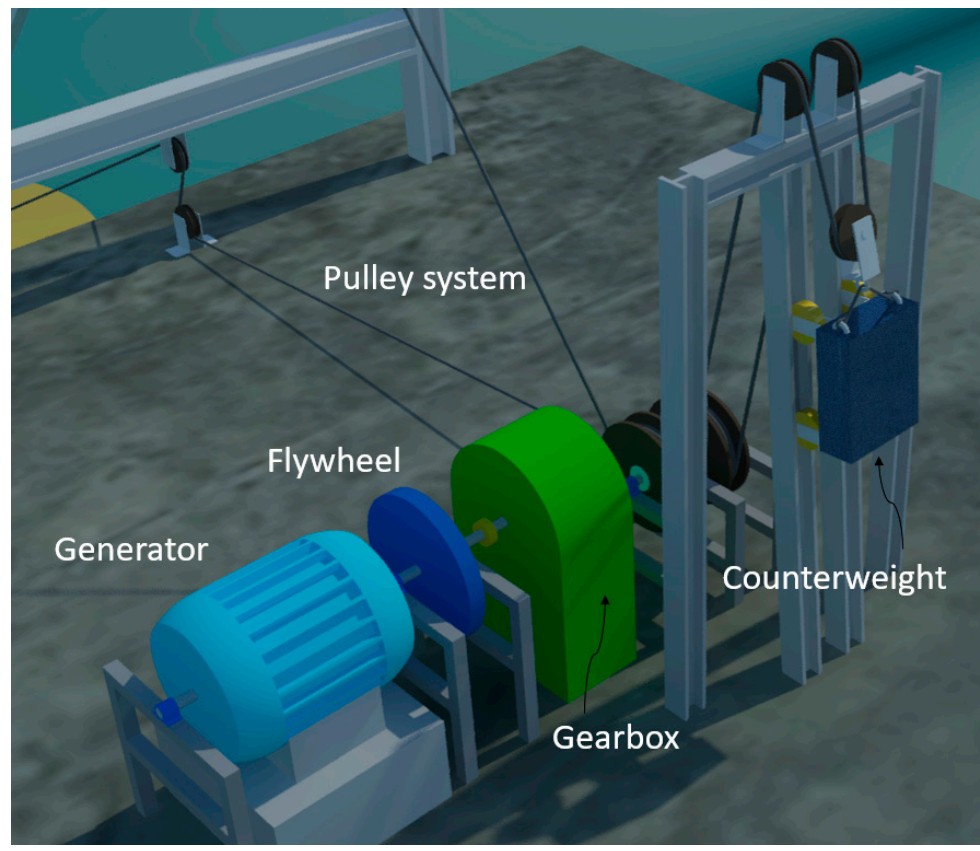

(**a**)

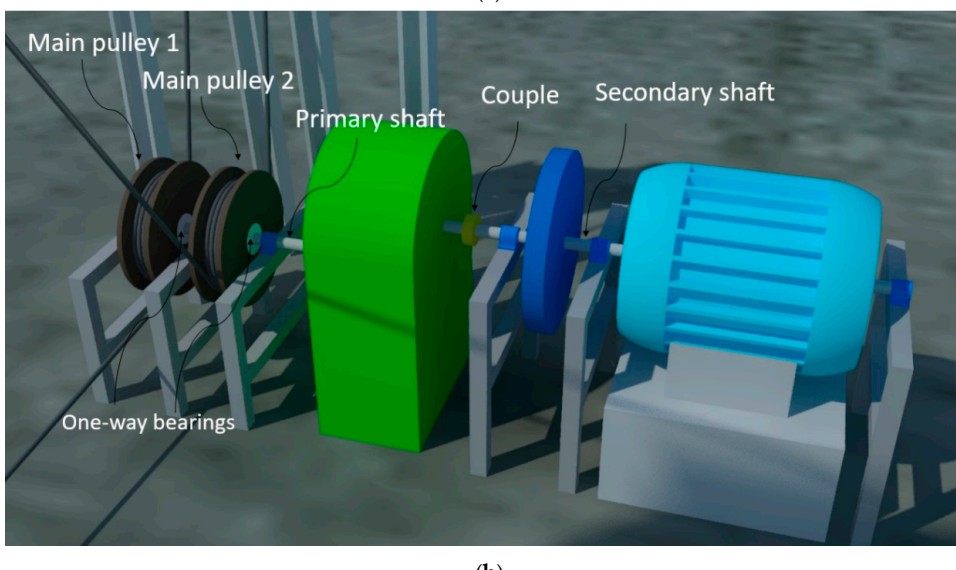

(**b**)

**Figure 2.** Direct mechanical drive PTO. (**a**) Back view. (**b**) Front view.

**Table 2.** Main parameters of the WEC.

| Parameter | Value |
| --- | --- |
| $h_1$ | 3.83 m |
| $h_2$ | 3.83 m |
| $h_3$ | 3.83 m |
| $R$ | 10 m |
| $m_0$ | 0.25 t |

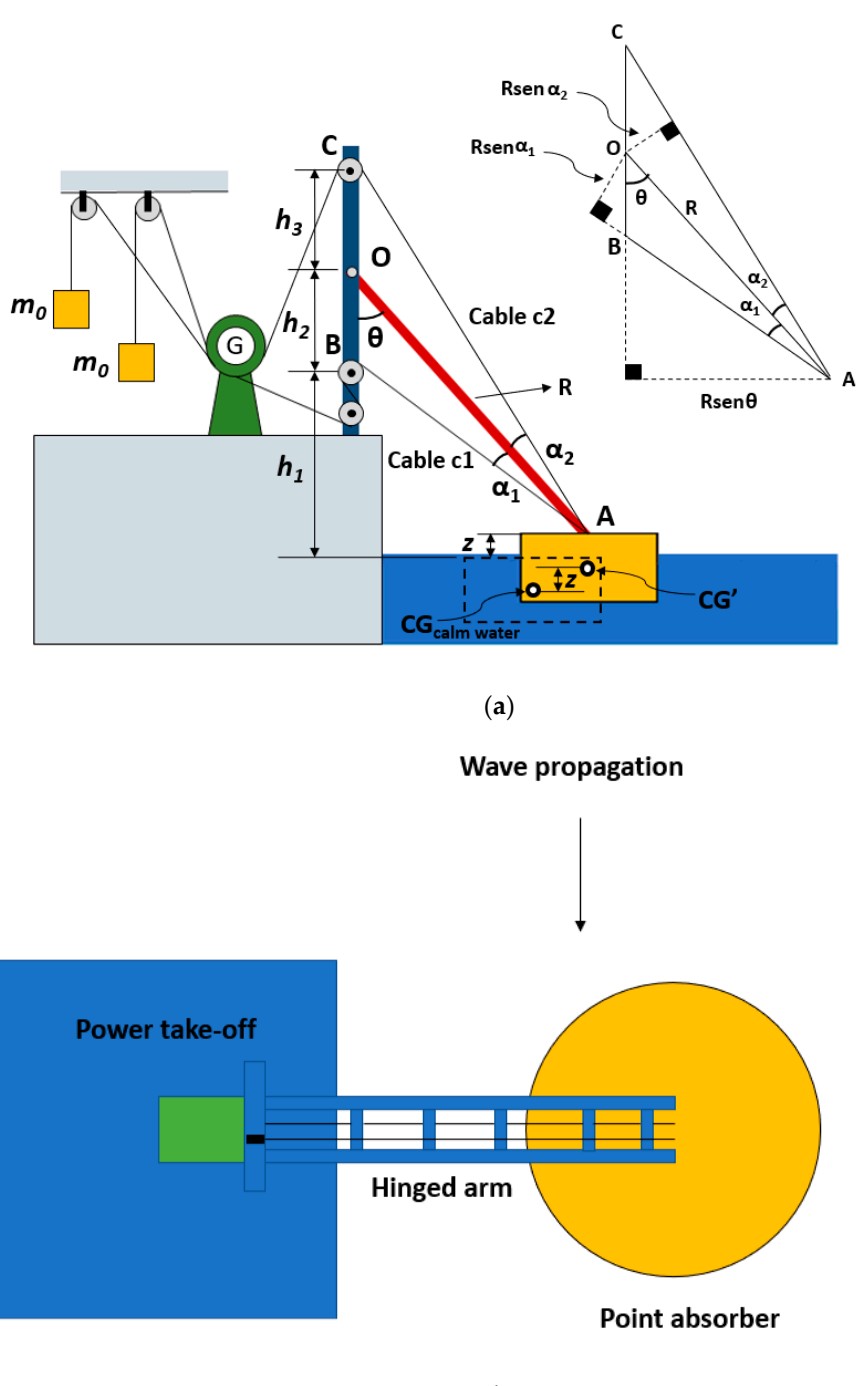

**Figure 3.** Schematic model of the wave energy converter. (**a**) Main dimensions of the WEC. (**b**) Wave propagation direction near the WEC.

*2.1. Dynamic Model of the WEC*

As a first approximation to the analysis of a new WEC, it is assumed that the point absorber is always in a vertical position and the weight of the articulated arm is neglected [37,38]. The equation of motion of the point absorber around the hinge O is defined by Equation (1).

$$-m_1 g R sin\theta + F_h R sin\theta - F_{c1} R sin\alpha_1 + F_{c2} R sin\alpha_2 = J\ddot{\theta} \tag{1}$$

where $J$ is the moment inertia of the center of gravity of the point absorber with respect to the point O; $\ddot{\theta}$ is the angular acceleration; $m_1$ is the mass of the point absorber; $g$ is the gravity acceleration; $F_{c1}$ and $F_{c2}$ are the cable tensions of cable c1 and cable c2, respectively; $F_h$ is the hydrodynamic force; R is the length of the hinged arm ($\overline{OA}$ distance). The variables $\alpha_1$ and $\alpha_2$ are described in Figure 3.

In Figure 3, the cable length $L_1$ between points A and B is defined in Equation (2). The relation of the time variation of the length $L_1$ to the angular velocity $\dot{\delta}_1$ of the main pulley 1 of radius $r$ is described in Equation (3). The cable length $L_2$ between points A and C is defined in Equation (4). The relation of the time variation of the length $L_2$ to the angular velocity $\dot{\delta}_2$ of the main pulley 2 of radius $r$ is described in Equation (5). The relation of the vertical projection of the hinged arm to the $z$ displacement of the point absorber is defined in Equation (6).

$$L_1 = \left[R^2 + h_2^2 - 2Rh_2cos\theta\right]^{1/2} \tag{2}$$

$$\dot{L}_1 = \dot{\delta}_1 r \tag{3}$$

$$L_2 = \left[R^2 + h_3^2 - 2Rh_3cos(\pi - \theta)\right]^{1/2} \tag{4}$$

$$\dot{L}_2 = \dot{\delta}_2 r \tag{5}$$

$$Rcos\theta = (h_1 + h_2 - z) \tag{6}$$

The linear potential flow theory is applied to calculate the hydrodynamic force. The direction of the wave considered in the present study is shown in Figure 3b; therefore, the oscillation of the point absorber is generated solely by the wave force in heave. The hydrodynamic force is defined in Equation (7).

$$F_h = F_b + F_r + F_e + F_v \tag{7}$$

where $F_b$ is the buoyance force, $F_r$ is the radiation force, $F_e$ is the excitation force and $F_v$ is the viscous damping force. The wave elevation is described by Equation (8).

$$z_w = A_w cos(\omega t) \tag{8}$$

where $A_w$ is the wave amplitude and $\omega$ is the wave angular frequency.

The gravity and buoyancy force define the restoring force $F_{rs}$, see Equations (9) and (10).

$$F_{rs} = F_b - m_1 g = -c_{33}z \tag{9}$$

$$c_{33} = \rho g A \tag{10}$$

where $\rho$ is the density of the water, $A$ is the cross section of the point absorber and $c_{33}$ is the stiffness coefficient.

$F_r$ is defined according to [39], see Equation (11), whereby the radiation convolution integral is solved by a direct method.

$$F_r = -a(\infty)\ddot{z} - \int_0^t K_r(t-\tau)\dot{z}(\tau)d\tau \tag{11}$$

where $K_r$ is the retardation function and $a(\infty)$ is the added mass. According to [40], $K_r$ can be calculated by Equation (12).

$$K_r(t) = \frac{2}{\pi}\int_0^\infty b_{33}(\omega)cos(\omega t)d\omega \tag{12}$$

where $b_{33}$ is the damping coefficient. The retardation function is shown in Figure 4, and the damping coefficient and the added mass of the point absorber are shown in Figure 5.

The excitation force $F_e$ is given by Equation (13).

$$F_e = \frac{H_w}{2}F_{33}cos(\omega t - \phi_{33}) \tag{13}$$

where $H_w$ is the wave height, $F_{33}$ is the wave force and $\phi_{33}$ is the phase shift. The last two parameters are obtained in Figure 6.

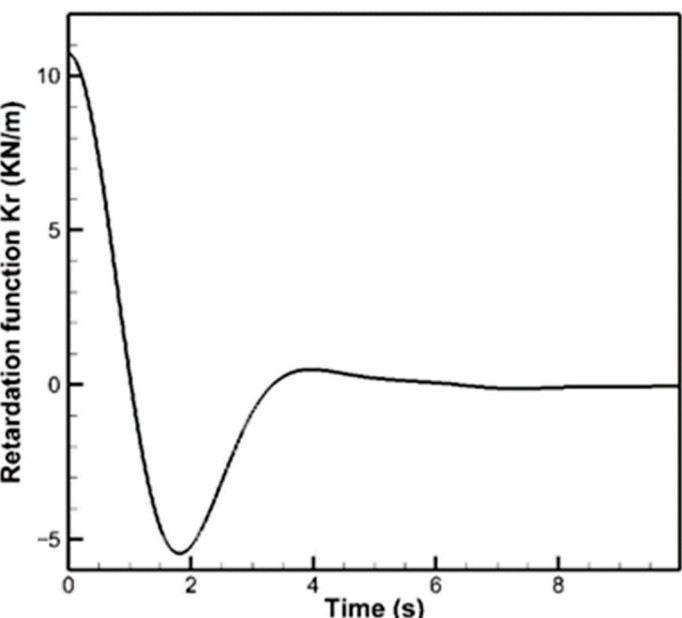

**Figure 4.** Retardation function $K_r$.

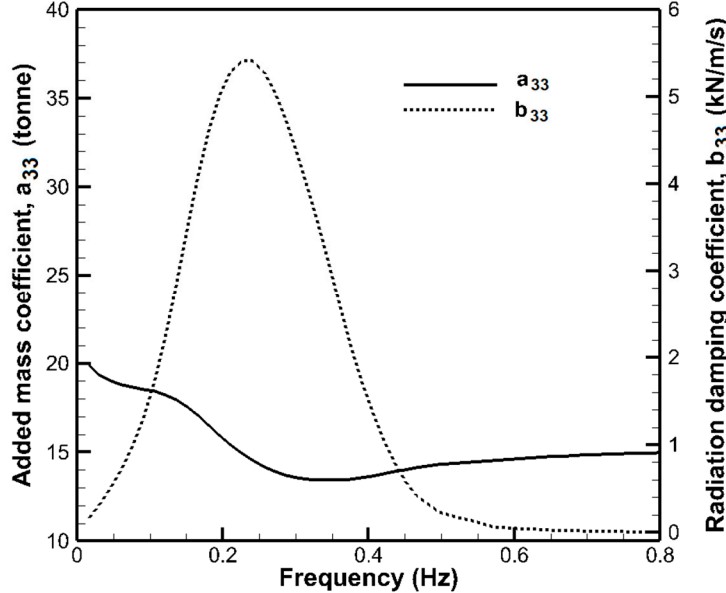

**Figure 5.** Added mass and radiation damping coefficients obtained via ANSYS/AQWA.

The viscous damping force $F_v$ is calculated based on the drag force described in [41], but rewritten according to [42], see Equation (14).

$$F_v = -\frac{1}{2}\rho C_d A(\dot{z} - u)|\dot{z} - u| \tag{14}$$

where $\rho$ is the water density, $C_d$ is the drag coefficient, $\dot{z}$ is the body velocity, $u$ is the undisturbed velocity of the fluid and $A$ is the cross-sectional area of the floating body. The value of the drag coefficient $C_d$ assumed in this study is 1.85; the drag coefficient of the point absorber considered has been calculated by [20].

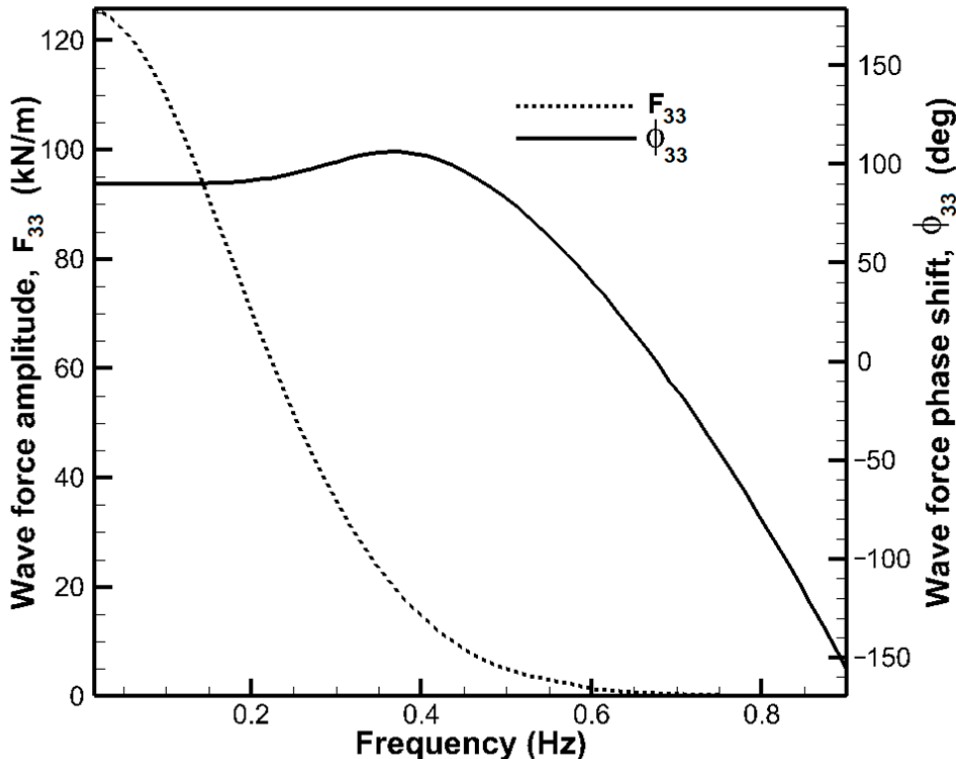

**Figure 6.** Wave force amplitude and wave force phase shift obtained via ANSYS/AQWA.

### 2.2. PTO System

The PTO system depends on the cable tensions $F_{c1}$ and $F_{c2}$, see Figure 3. The cable wraps around each of the main pulleys and transmits torque to the shaft; it is assumed that the cable does not slip on the main pulleys. The operation of the PTO system is defined by the coupling and uncoupling of the point absorber and the electric generator. Considering the coupling of the PTO in the upward movement corresponding to the main pulley 1 (the main pulley 2 rotates freely), and neglecting the torque generated by the support bearings of the primary shaft (see Figure 7), the equations of motion of the main pulley 1 and the counterweight according to the free body diagram in Figure 8 are defined by Equations (15) and (16), respectively.

$$\left[J_1 + c^2 J_2\right]\ddot{\delta}_1 = F_{c1}r - F_{c01}r - \left[c\left(T_g + 3T_b\right)\right] \tag{15}$$

$$m_0\ddot{\delta}_1 r = F_{c01} - m_0 g \tag{16}$$

The expression in brackets on the left side of Equation (15) is the equivalent inertia referred to the primary shaft (low-speed shaft). $J_1$ corresponds to the sum of the inertia of the main pulley ($J_p$), the inertia of the primary shaft ($J_{s1}$) and the inertia of the gearbox ($J_{gb}$). $J_2$ corresponds to the sum of the inertia of the generator ($J_g$), the inertia of the secondary shaft ($J_{s2}$) and the inertia of the flywheel ($J_f$). $\ddot{\delta}_1$ is the angular acceleration of the main pulley 1, $r$ is the radio of the main pulley 1, $c$ is the gear ratio (high angular velocity/low angular velocity), $T_b$ is the torque of each secondary shaft bearing, $T_g$ is the torque of the generator and $F_{c01}$ is the cable tension between the main pulley 1 and the counterweight. If the point absorber and the electric generator are uncoupled, Equation (15) is rewritten as Equation (17).

$$J_1\ddot{\delta}_1 = F_{c1}r - F_{c01}r \tag{17}$$

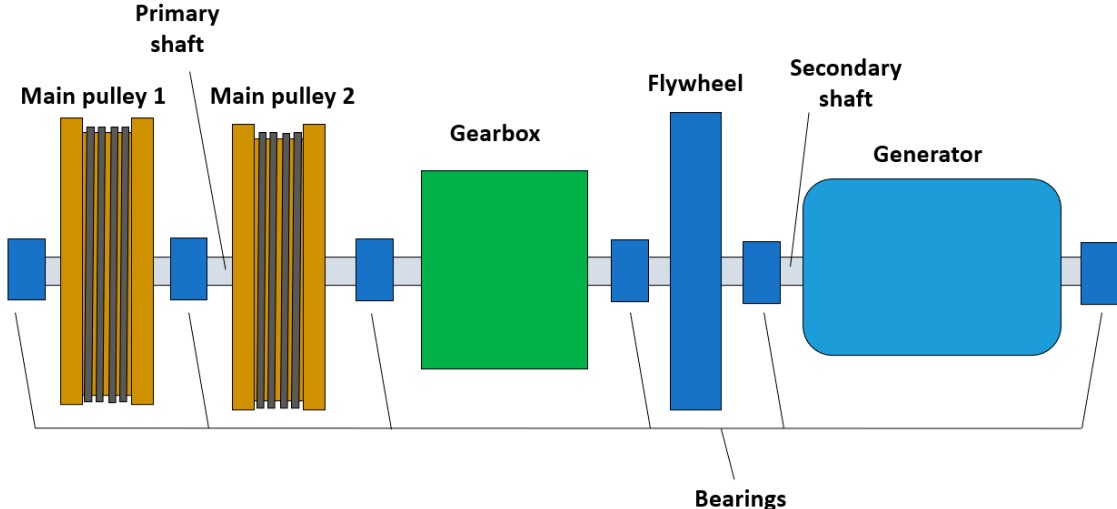

**Figure 7.** PTO system.

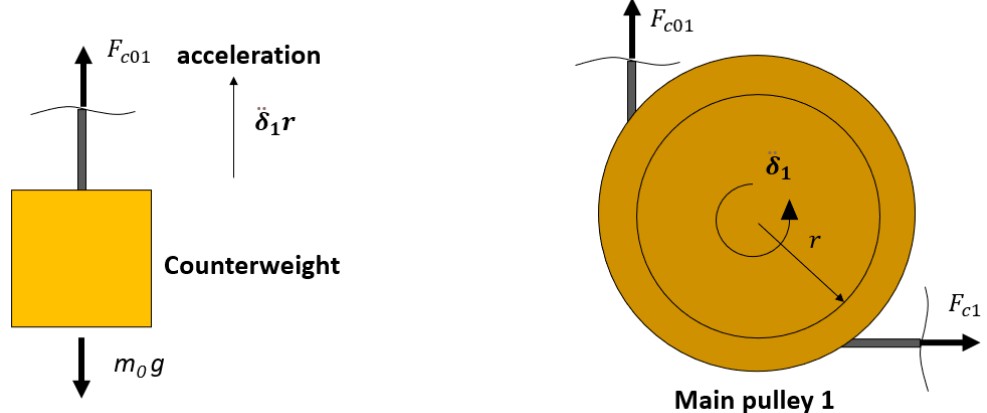

**Figure 8.** Diagram of the free body of the main pulley 1 and the counterweight.

Using the same analysis when the WEC captures energy only in the downward movement corresponding to the main pulley 2 (the main pulley 1 rotates freely), the equation of motion of the main pulley 2 for the coupling and uncoupling of the PTO and the counterweight are shown in Equations (18)–(20), respectively.

$$\left[ J_1 + c^2 J_2 \right] \ddot{\delta}_2 = F_{c2}r - F_{c02}r - \left[ c \left( T_g + 3T_b \right) \right] \tag{18}$$

$$J_1 \ddot{\delta}_2 = F_{c2}r - F_{c02}r \tag{19}$$

$$m_0 \ddot{\delta}_2 r = F_{c02} - m_0 g \tag{20}$$

where $\ddot{\delta}_2$ is the angular acceleration of the main pulley 2 and $F_{c02}$ is the cable tension between the main pulley 2 and the counterweight.

Two electric generators of low and high power are considered to analyze the performance of the WEC. The generators are permanent magnets [43], and the characteristics of the electric generators are shown in Table 3 while the power curves are shown in Figure 9. The generator torque without MPPT is calculated by Equation (21).

$$T_g = P_g / (\eta_g \omega_g) \tag{21}$$

where $\eta_g$, $P_g$ and $\omega_g$ are the efficiency, power and speed of the generator.

**Table 3.** Parameters of the electric generators.

| Parameters | G1 | G2 |
|---|---|---|
| Rated power (W) | 22,678 | 95,484 |
| Rated speed (rpm) | 400 | 350 |
| Rotor inertia (kgm$^2$) | 1.270 | 7.620 |
| Rated torque (Nm) | 611 | 2771 |
| Efficiency, $\eta_g$ (%) | 89 | 94 |
| Phase resistance (Ohm) at 20 °C | 0.2 | 0.02 |
| Phase inductance (mH) | 0.94 | 0.27 |
| Voltage at no load (back emf) at 20 °C | 299 | 336 |

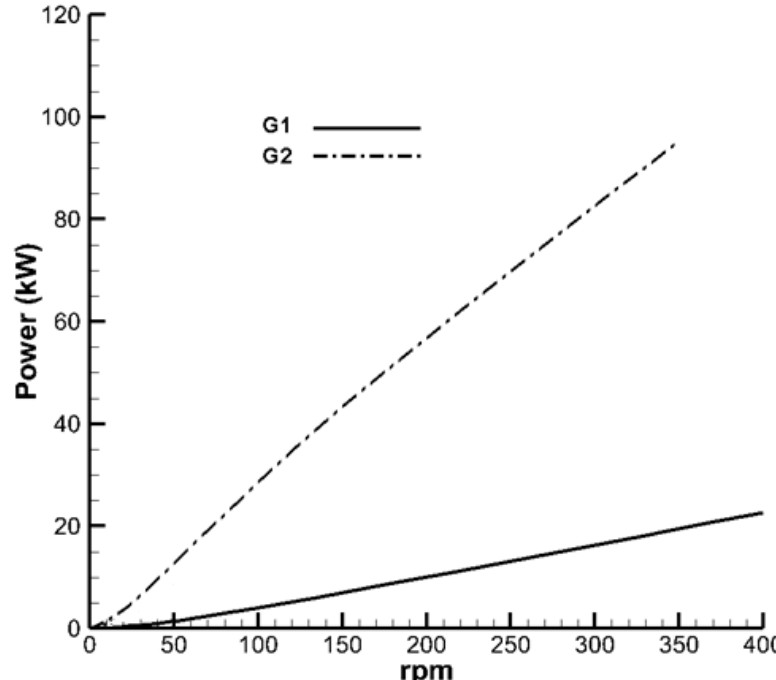

**Figure 9.** Power curves of the generators.

The torque $T_b$ is approached using Petroff's method of lubrication analysis [44], see Equation (22).

$$T_b = \frac{4\pi^2 r_s^3 l \mu N}{c_s} \tag{22}$$

where $r_s$ is the shaft radius, $l$ is the bearing length, $\mu$ is the dynamic viscosity of the lubricator, $N$ is the shaft speed and $c_s$ is the radial clearance.

The coupling of the PTO system via the main pulley 1 is described by Equation (23). When the output shaft speed $c\dot{\delta}_1$ and the generator speed $\omega_g$ satisfy this equation, the generator and the point absorber are coupled, then the generator shaft speed takes the value of $c\dot{\delta}_1$. Similarly, for the main pulley 2, coupling occurs when the output shaft speed $c\dot{\delta}_2$ and $\omega_g$ satisfy Equation (24).

$$\omega_g < c\dot{\delta}_1 \tag{23}$$

$$\omega_g < c\dot{\delta}_2 \tag{24}$$

If the values of Equation (23) or Equation (24) are not met, the generator and point absorber are uncoupled. Then, the speed of generator is defined by Equation (25), if

the generator is within its operating speed range (see Figure 7), otherwise it is given by Equation (26).

$$\omega_g(t + \Delta t) = \omega_g(t) - \frac{(T_g + 3T_b)}{J_2}\Delta t \tag{25}$$

$$\omega_g(t + \Delta t) = \omega_g(t) - \frac{3T_b}{J_2}\Delta t \tag{26}$$

where $t$ and $\Delta t$ are the time and time step variables of the numerical simulation.

The power is obtained from the generator or the flywheel kinetic energy; the flywheel inertia considered in the present study is 40 kgm$^2$. The generator runs only within its operating speed range; therefore, in Equations (15) and (18), $T_g$ depends on the generator shaft speed. When the generator exceeds its maximum speed, the torque $T_b$ of the bearings reduces the generator shaft speed, see Equation (26).

The mean power $P_m$ is determined by Equation (27).

$$P_m = \frac{1}{T}\int_0^T P_g \, dt \tag{27}$$

where $T$ represents the last 10 periods of the numerical simulation and $P_g$ is the instantaneous power of the generator. The numerical simulation time considers 10,000 s; the first 1000 s are discarded to avoid transient instabilities.

The capture width ratio of the WEC is calculated by Equation (28).

$$CWR = \frac{P_m}{P} \tag{28}$$

where $P$ is the incident wave power defined by Equation (29).

$$P = \left[\frac{1}{8}\rho g H_w{}^2\right]\left[\frac{c}{2}\left(1 + \frac{2kd}{sinh(2kd)}\right)\right]D \tag{29}$$

where the first factor is the mean wave-energy density per unit horizontal area, the second factor is the group velocity [45], $c$ is the phase velocity, $k$ is the wave number, $d$ is the water depth and $D$ is the diameter of the floating body.

## 3. Maximum Power Point Tracking

Most WECs based on rotating generators are considered to be permanent magnet synchronous generators (PMSGs); thus, in this work, a generic model of a PMSG using the P&O MPPT method is considered in order to compute the torque according to Equations (30) and (31), where $P_g$ is the output electrical power; $K$ is the back electromotive force constant of the generator in vs/rad; $N_p$ is the pole pairs number; $L_s$ and $R_s$ are the stator inductance and resistance, respectively; $\eta_g$ is the efficiency of the generator and $R_g$ is the generator resistance, which is the parameter controlled by the MPPT algorithm.

$$P_g = \frac{K^2\omega_g^2 R_g}{2\left((N_p\omega_g L_s)^2 + (R_s + R_g)^2\right)} \tag{30}$$

$$T_g = \frac{P_g}{\eta_g\omega_g} = \frac{K^2\omega_g R_g}{2\left((N_p\omega_g L_s)^2 + (R_s + R_g)^2\right)} \tag{31}$$

The MPPT P&O algorithm is performed to extract the maximum available power by periodically adjusting the $R_g$ value, thus modifying the generator torque under a gradient-ascent approach as detailed in [33]. The control of $R_g$ is realized by means of a suitable electronic converter, in this way the electric side of the WEC acts like a variable resistor. A flowchart of the P&O MPPT method considered in this work is presented in Figure 10. Variations in $R_g$ are discrete with the $R_{step}$ value, after each variation in the mean power $P_g$ is computed and compared to the previous value to set the next value of $R_g$. Since the tracking of the maximum power is limited by the generator-rated value, the maximum

speed and the extreme allowable value for RG ($R_{min}$ and $R_{max}$), the algorithm considers these constraints.

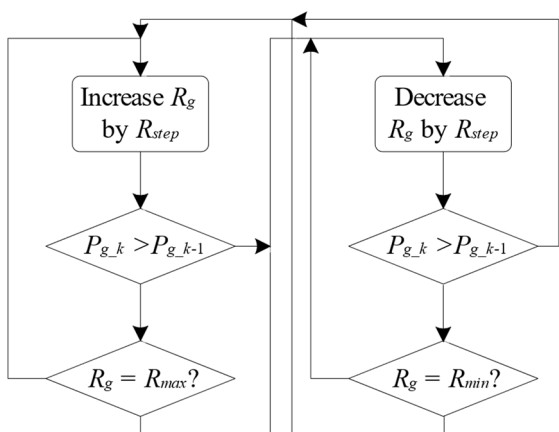

**Figure 10.** Flowchart of the considered perturb and observe MPPT method.

## 4. Simulation Results

The simulation results are presented in two sections. First, an analysis of the influence of different parameters, such as the gear ratio, wave period and height are presented; this is conducted based on two generators (G1 and G2), for which the technical details are listed in Table 3. Second, the simulation results of the time series of the main variables are presented for certain operational parameters with generator G1.

### 4.1. WEC Performance: Influence of Key Parameters

Under defined $H_w$ and $T_p$ conditions, operating with a higher gear ratio (GR) implies a higher generator speed. Since the rotor speed is variable, there are time intervals of maximum speed. As the GR is increased, these intervals of maximum speed can mean operating above the generator speed limit. This operation is undesirable; thus, in these conditions, the system disconnects the generator output, resulting in zero generation. There is a critical value of GR above which intervals of zero generation begin to occur. As the GR increases over this critical value, the intervals of zero generation become longer, thus reducing the average generated power until it reaches zero. This is noticeable in Figures 11–14, which present the mean power vs. the GR for generators G1 and G2 under different conditions of wave period and height. It is also noticeable that a higher incident power (i.e., higher $H_w$ and lower $T_p$) leads to a lower critical value of GR. This is predictable, since higher power is related to higher generator speed; thus, a lower GR is expected.

When operating with MPPT, the value of Rg is adjusted to increase both the speed and the torque, aiming to extract the maximum available power. Thus, the critical value of GR is expected to be less than or equal to operation without MPPT. Moreover, for a GR lower than the critical value, operation with MPPT should produce a higher mean power than without MPPT. For GR values above the critical value, MPPT operation is expected to produce the same power (or even a bit lower) as without MPPT. Since for both operation conditions (with or without MPPT) the incident power is the same, the CWR for MPPT operation is also expected to be higher. On the other hand, for a GR above the critical value, the MPPT method is disabled, and the converter operates the same as without MPPT.

For the G2 generator without MPPT, the power takes the value of zero in some cases. This behavior occurs because the operating speed of the generator in a steady state exceeds the nominal speed of the generator; therefore, there is no power generation. This behavior is removed when the WEC operates with MPPT.

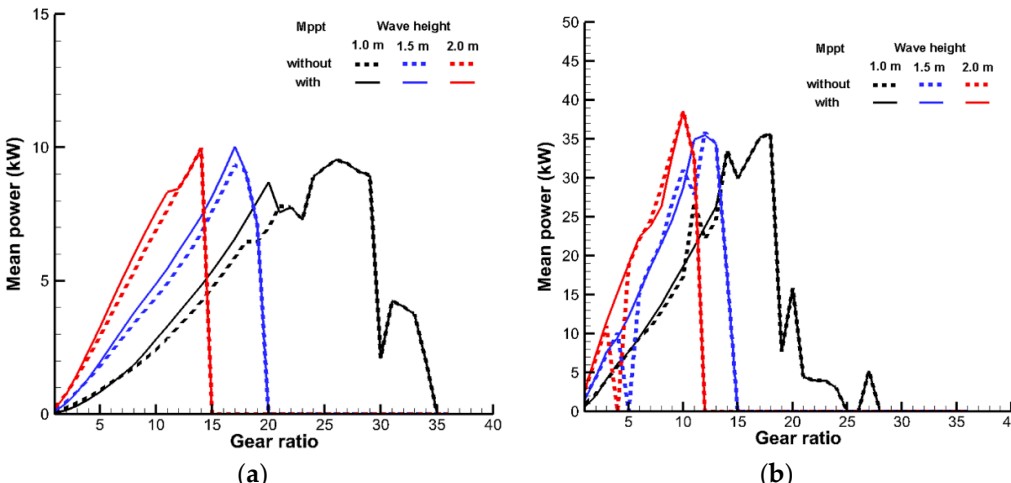

**Figure 11.** Mean power vs. GR at different wave heights and at wave period Tp =12 s: (**a**) generator G1, (**b**) generator G2.

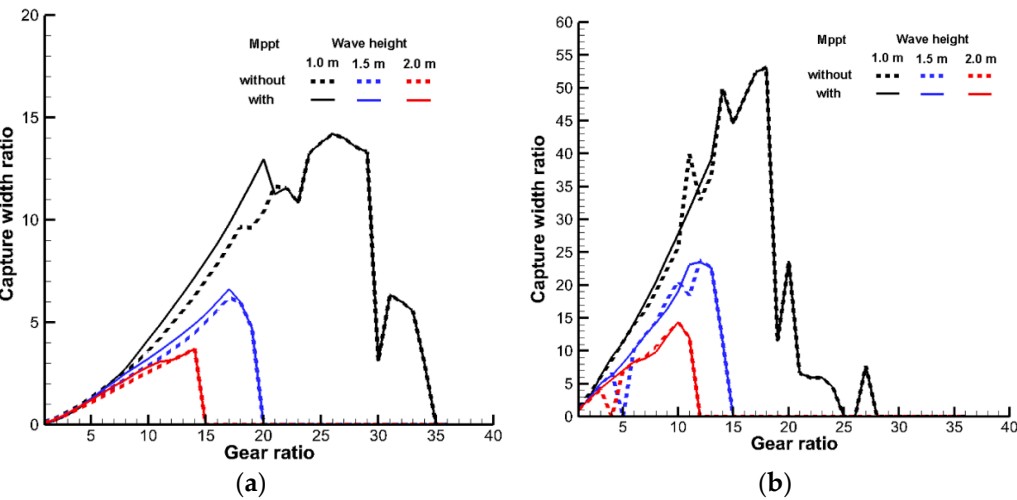

**Figure 12.** CWR vs. GR at different wave heights and at wave period Tp = 12 s: (**a**) generator G1, (**b**) generator G2.

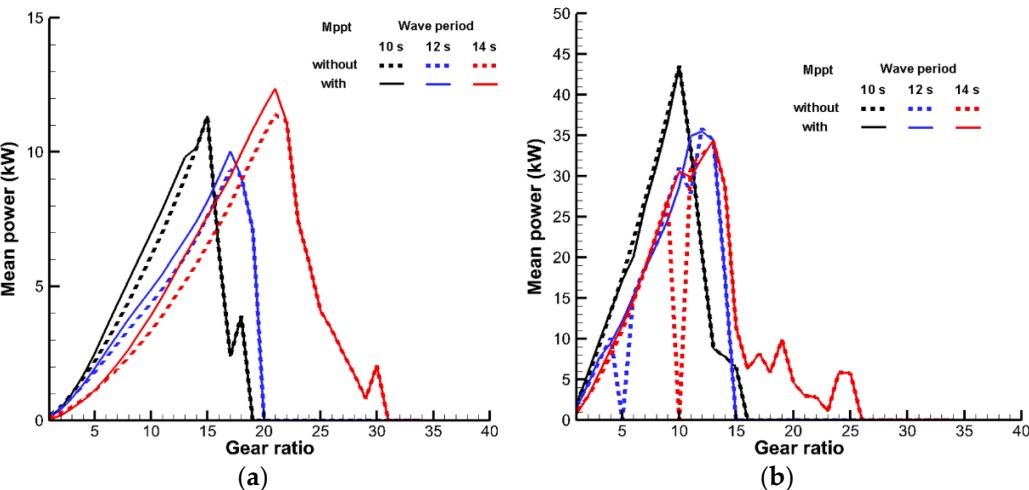

**Figure 13.** Mean power vs. GR at different wave periods and at wave height Hw = 1.5 m: (**a**) generator G1, (**b**) generator G2.

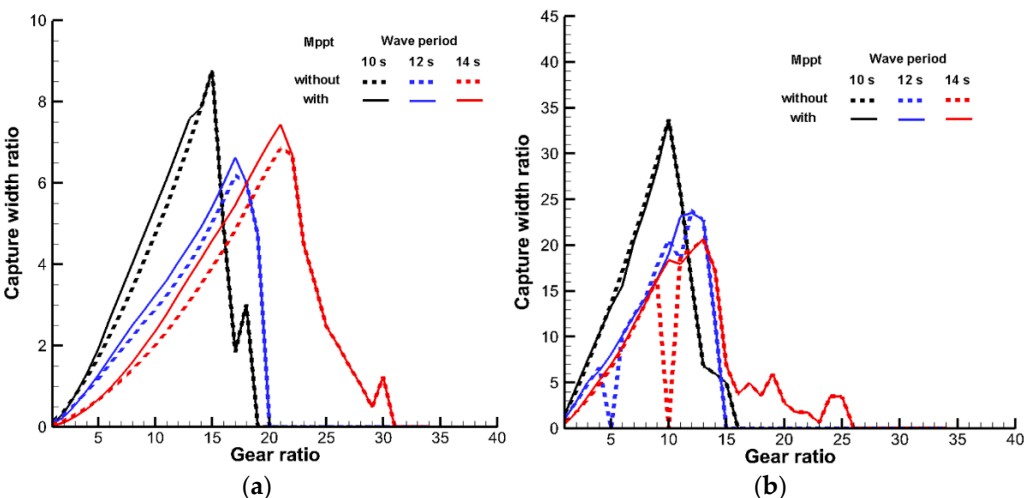

**Figure 14.** CWR vs. GR at different wave periods and at wave height Hw = 1.5 m: (**a**) generator G1, (**b**) generator G2.

From Figures 11 and 13 it is noticeable that, for GR values lower than the critical value, the proposed WECs produce more power when operating under the MPPT algorithm. Above the critical GR, the power for operation with MPPT is almost equal to that without MPPT. These are the expected results. For the best operating conditions of the two generators, the maximum achieved value in terms of mean power is about 50% of the rated power according to the datasheet: 44 kW for G2 and 12 kW for G1, both for Hw = 1.5 and Tp = 10 s. Moreover, the CWR is also higher for operation with MPPT, as can be seen in Figures 12 and 14, achieving values of up to 14% for G1 and 53% for G2 when operating with Tp = 12 s and Hw = 1 m. These results may lead one to think that G2 is the most suitable generator for the WEC. However, the results obtained with generator G1 are quite promising, particularly considering that operating with smaller generators (up to 25 kW) represents a significant opportunity as this implies a substantial reduction in costs. Furthermore, the results confirm that the inclusion of the MPPT method allows the WECs to achieve the same mean power with a lower GR than without MPPT. This means lower losses and higher efficiency.

### 4.2. Time Series for Generator 1

The conditions selected for the time series analysis are: Hw = 1 m, Tp = 12 s and GR = 20. The main waveforms are presented in Figure 15 for steady-state operation (last 60 s of simulation). Results are presented for both operation without MPPT (left) and with MPPT (right). The selected variables are: displacement and velocity of the point absorber, wave excitation force (upper), instantaneous torque and power (middle) and generator speed (lower).

For operation without MPPT, the generator is loaded with a constant impedance, which produces a torque proportional to the speed; in time, the power is proportional to the square of the speed. On the other hand, when operating under the MPPT method, the torque is always higher than without MPPT; the algorithm increases the torque to extract more power. The limit is the rated torque (611 Nm). For both conditions it is noticeable that a minimum generator speed is achieved for maximum displacement, which also corresponds to the minimum torque. However, with MPPT, the torque around the minimum displacement is at its maximum; in this way, operating with MPPT produces more power.

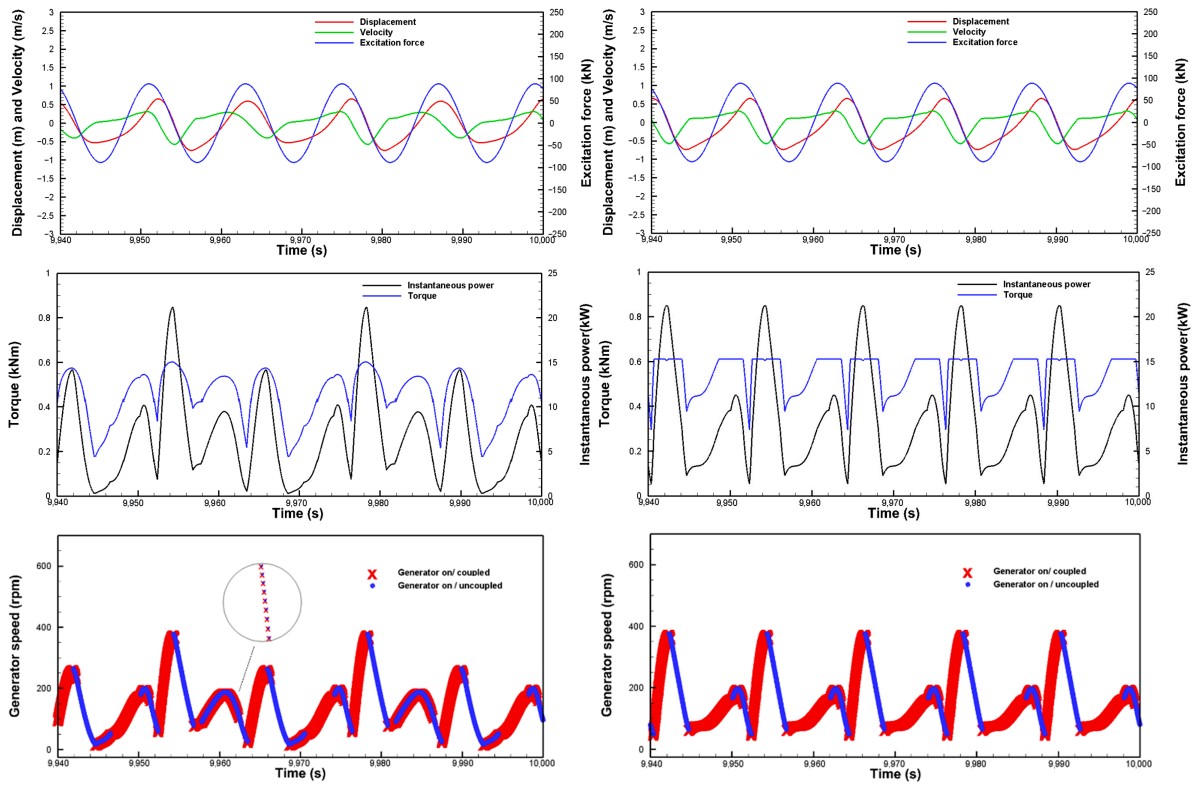

**Figure 15.** Main waveforms for G1 operation without MPPT.

MPPT Optimization of Rg

As detailed, the MPPT algorithm aims to maximize the power generation by means of adjusting the load resistance seen by the generator. Figure 16 shows the time response of the MPPT algorithm. For the first 300 periods an increase in Rg leads to an increase in the mean power. When Rg reaches the value of 0.32 Ω, an increase in Rg results in a decrease rather than an increase in power, thus the MPPT has been achieved. The MPPT algorithm keeps the Rg value oscillating around 0.32 Ω. Therefore, the mean power also oscillates around its maximum value.

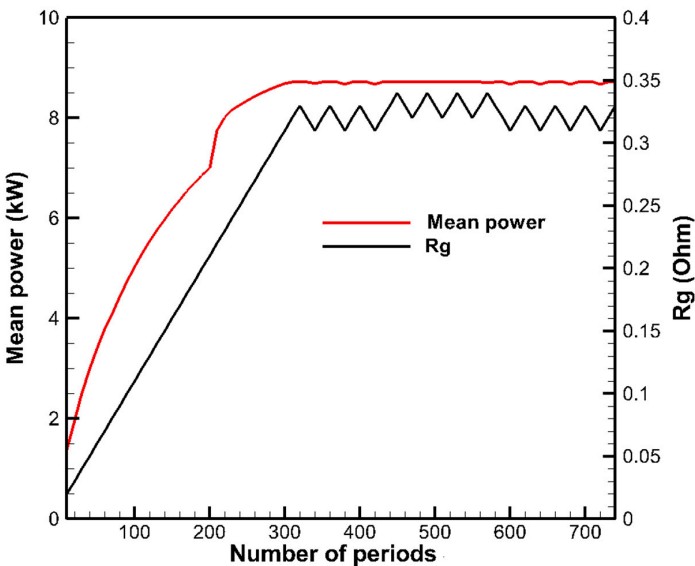

**Figure 16.** MPPT optimization of the generator load.

## 5. Conclusions

The influence of the perturb and observe MPPT method applied to a new wave energy converter based on a point absorber, a hinged arm and a direct mechanical drive PTO system is studied herein. The PTO system is composed of a pulley system, a counterweight, a gearbox, a flywheel and an electric generator. The analysis of the WEC performance in regular waves considers two electric generators with a rated power of 22.7 kW and 95.5 kW, respectively. In general, the electrical power generation of the WEC increases with MPPT for lower GR values until a critical GR value is reached, after which the power decreases. The contribution of MPPT is higher for the low-power generator. The maximum mean power achieved is about 50% of the rated power for both generators. The high CWR values of the WEC occur at low values of period and wave height; the maximum value obtained is 53%, corresponding to the high-power generator G2. However, the low-power generator G1 shows a better response with the inclusion of MPPT; that is, having the same mean power with less GR than without MPPT, and smaller generators mean a substantial reduction in costs. In future work, the power performance of the WEC and the contribution of MPPT in irregular waves will be analyzed.

**Author Contributions:** Methodology, J.C.U.P. and G.O.G.A.; Software, C.L.M.R. and G.O.G.A.; Formal analysis, J.C.U.P.; Writing – original draft, G.O.G.A.; Funding acquisition, G.O.G.A. All authors have read and agreed to the published version of the manuscript.

**Funding:** This research was funded by the Universidad Tecnológica del Perú (resolución rectoral no. 0047-2022-R-UTP).

**Institutional Review Board Statement:** Not applicable.

**Informed Consent Statement:** Not applicable.

**Data Availability Statement:** Data is contained within the article.

**Conflicts of Interest:** All authors declare that they have no conflict of interest.

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
