# Peer review of "Study of a New Wave Energy Converter with Perturb and Observe Maximum Power Point Tracking Method"

_sustainability, doi:10.3390/su151310447_

Round 1

Reviewer 1 Report

The work done falls within the framework of renewable energy, as it is an important work. Among the notes I mention:

1. Do not use abbreviations in keywords.

2. Do not use this writing for references [9]-[14], as it is better to write like this [9,14]. Please follow the journal's policy in writing the paper.

3. Please, write the text in one format

4. In the References section, you must write the references according to the journal's policy.

5. The goal of the work is to generate electrical energy from waves??? The exact idea is not clear.

6. What type of generator used? its characteristics?

7. What are the limitations of the work performed?

8. Can this idea be relied upon in the case of high waves?????

9. What about the durability of the system?????.

Author Response

Reviewer #1:

The work done falls within the framework of renewable energy, as it is an important work. Among the notes I mention.

R: We would like to thank reviewer for the comments and efforts to improve the papers quality.

  • Do not use abbreviations in keywords.

R: The authors apologize for this mistake; all abbreviations have been removed from keywords.

  • Do not use this writing for references [9]-[14], as it is better to write like this [9,14]. Please follow the journal's policy in writing the paper.

R: The authors apologize for the misunderstanding. Our purpose was to cite a total of 5 papers (from reference 9 to reference 14), the suggestion of the reviewer is intended for the citation of only two papers. Nevertheless, in the revised version, the citation style has been modified according with the journal’s policy.

  • Please, write the text in one format.

R: Text is now written in only one format according to template.

  • In the References section, you must write the references according to the journal's policy

R: The references section has been modified in accordance with the journal's policy

  • The goal of the work is to generate electrical energy from waves? The exact idea is not clear

R: We really apologize for the lack of clearness in such an important aspect. The goal of the work is to evaluate the performance of the proposed WEC under the influence of the Perturb and Observe algorithm for Maximum Power Point Tracking from the perspective of electric power generation. The abstract has been modified including the following sentence:

“This study aims to address the analysis of the influence of the Perturb and Observe method for MPPT in the electrical power performance...”

  • What type of generator used? its characteristics?

R: The manuscript considers Permanent Magnet Synchronous Generators. In the reviewed version this is mentioned in the paragraph before Table 3, which also list the main parameters of the considered generators based on commercially available models as detailed in reference [43].

  • What are the limitations of the work performed?

R: In its current status, this research has two main aspects to be improved in future works. A first aspect is the consideration of only the vertical component of the hydrodynamic force that excites the point absorber, the direction of the considered wave (see Fig.3b) allows this approximation. Future works may consider the effects of horizontal hydrodynamic force. A second aspect to be analyzed in future works is the behavior of the proposed WEC in front of irregular waves, since the present manuscript is limited to regular waves.

  • Can this idea be relied upon in the case of high waves?

R: According to the results obtained, the power generation increases with the wave height, however, the capture width ratio decreases. Also, it indicates the power performance of the WEC is bettter at low values of period and wave height. Therefore, the operation of WEC in high waves would result in lower efficiency. This work represents a first analysis of the operation of this new WEC, the analysis of the behavior of the WEC at different sea states will be conducted in the future.

  • What about the durability of the system?

R: The present study does not analyze the durability of the system. WEC maintenance is an important factor in the life cycle of these devices. According to reference [15], WECs based on an absorption point connected to ground through an articulated arm are gaining interest due to their easy installation and maintenance; this represents a point in favor. Dalton et al. 2014 [A] showed that the Wavestar device, which is similar to the proposed WEC, has a life cycle cost comparable to offshore wind, but much higher than onshore wind. This may represent a gross value of the life cycle of the proposed device.

_______________________________________________________________________________

Reference:

[A] G. Dalton, D. Madden and M. C. Daly, "Life Cycle Assessment of the Wavestar," 2014 Ninth International Conference on Ecological Vehicles and Renewable Energies (EVER), Monte-Carlo, Monaco, 2014, pp. 1-9, doi: 10.1109/EVER.2014.6844034.

Reviewer 2 Report

In this paper, the authors address the analysis of the influence of the perturb and observe MPPT in the power performance of a WEC composed of a point absorber, a hinged arm and a direct mechanical drive PTO system. The PTO is characterized by a pulley system, one-way bearings, a gearbox, a flywheel and an electric generator, in the present study it is considered a cylindrical point absorber.

The topic this paper is so interesting, and the results obtained are correct. Therefore, the publication of this paper is recommended after the following minor revision.

1- The novelty and contribution of the paper should be better highlighted in the introduction.

2- What are the key features of the proposed technique? (properties, characteristics, and weaknesses).

3- The authors discussed the new wave energy converter with perturb and observe maximum power point tracking method, but the references are largely restrictive to a specific domain of energy. The authors should give a review of a broader scope and refer to renowned studies or textbooks in the relevant area. For this purpose the authors can refer to the following techniques:

https://doi.org/10.1063/5.0141228; https://doi.org/10.3390/fractalfract6120745;... etc.

4-  The significance of the obtained results should be highlighted.

 5- Can the authors explain what kind of difficulties have been met when perform the experimental tests?

 6-  In conclusion authors need to add more information about future direction.

In short, the results of the paper are very satisfactory. The findings are clear and concise. After correction along the above lines, the paper can be accepted for publication.

Author Response

Reviewer #2:

In this paper, the authors address the analysis of the influence of the perturb and observe MPPT in the power performance of a WEC composed of a point absorber, a hinged arm and a direct mechanical drive PTO system. The PTO is characterized by a pulley system, one-way bearings, a gearbox, a flywheel and an electric generator, in the present study it is considered a cylindrical point absorber.

The topic this paper is so interesting, and the results obtained are correct. Therefore, the publication of this paper is recommended after the following minor revision.

R: We would like to thank reviewer for the comments and suggestions to improve the quality of the paper.

  • The novelty and contribution of the paper should be better highlighted in the introduction.

R: We apologize for the lack of emphasis in these two major aspects of the work. In the revised version of the manuscript includes the introduction has been modified to highlight both the contribution and novelty of the work.

  • What are the key features of the proposed technique? (properties, characteristics, and weaknesses).

R: The proposed technique demonstrates a simplification of the wave-to-wire model for the novel Wave Energy Converter (WEC), particularly concerning the hydrodynamic force. In this study, only the vertical hydrodynamic force is taken into account as the sole contributor to the excitation of the point absorber's movement. This approach is justified based on the assumed direction of the waves in the current investigation. However, for waves propagating in other directions, a redefinition of the hydrodynamic force becomes necessary. Furthermore, certain assumptions are made, namely the neglect of the articulated arm's weight and the vertical displacement of the point absorber. Although these assumptions are not entirely accurate, they serve as a reasonable initial approximation, as indicated by previous works [37-38]. Despite the simplification of the wave to wire model developed, it shows interesting results of the power performance of the WEC, different variables can be analyzed over time; in the present study the most relevant variables are considered.

  • The authors discussed the new wave energy converter with perturb and observe maximum power point tracking method, but the references are largely restrictive to a specific domain of energy. The authors should give a review of a broader scope and refer to renowned studies or textbooks in the relevant area. For this purpose, the authors can refer to the following techniques.

R: We really appreciate your suggestion. The present work belongs to the area of oceanic renewable energies, for which reason one of his suggestions was included in the manuscript. In the revised version, the introduction includes more references related to recent studies.

  • The significance of the obtained results should be highlighted.

R: The main results have been highlighted in the abstract.

  • Can the authors explain what kind of difficulties have been met when perform the experimental tests?

R: In the current status of the research, no experimental test has been performed yet. Nevertheless, some aspects have been identified as potential difficulties for that future stage: The pulley system must be placed with high precision to transmit torque to the main pulleys 1 and 2, slipping between the cable and the main pulleys must be avoided. The cable must be taut at all times.

  • In conclusion authors need to add more information about future direction.

R: Authors really appreciate the suggestions of the reviewer to improve the quality of the paper. In the revised version, future works are commented in the final section.

  • In short, the results of the paper are very satisfactory. The findings are clear and concise. After correction along the above lines, the paper can be accepted for publication.

R: We would like to thank reviewer for the opinion about this research work and the suggestions to improve the papers quality

Reviewer 3 Report

I have the following comments:

1.   Expand WEC in paragraph 1. All abbreviations, no matter how trivial, should be expanded at the first usage within the manuscript (not including abstract)

2.   I do not understand “[3] show 31 developers from all over the world use the direct mechanical drive system.”

3.   The introduction is very short and weak. Please expand it, and include more recent references.

4.   What is PTO (Expand)

5.   Table 1 – please correct the spelling

6.   Table 2 – Please change the title. Otherwise “mass” cannot be included in dimensions

7.   Eq .1 – what is sen?

8.   Table 3 – the “k” in kg-m2 should be lower case

9.   Please add a reference for Eq. 27 – how did you obtain it?

10.                Figure 10 is not clear

Moderate editing of English language required

Author Response

Reviewer #3:

R: We would like to thank reviewer for the comments and suggestions to improve the quality of the paper.

  • Expand WEC in paragraph 1. All abbreviations, no matter how trivial, should be expanded at the first usage within the manuscript (not including abstract).

R: In the revised version of the manuscript all abbreviations are first expanded at the first appearance in the manuscript starting from the introduction.

  • I do not understand “[3] show 31 developers from all over the world use the direct mechanical drive system.”

R: We apologize for the lack of clearness of this sentence. The text has been modified in the revised version to emphasize that the PTOs based on direct mechanical drive are very well-known and common, as reported in [3].

  • The introduction is very short and weak. Please expand it, and include more recent references.

R: We really appreciate your suggestion. In the revised version of the manuscript introduction has been improved by including more precise ideas and more recent references.

  • What is PTO (Expand).

R: PTO is the abbreviation of Power-take-off. In the revised version of the manuscript PTO is expanded within the introduction section.

  • Table 1 – please correct the spelling

R: It has been corrected.

  • Table 2 – Please change the title. Otherwise “mass” cannot be included in dimensions.

R: The word has been changed by parameters.

  • Eq .1 – what is sen?.

R: We apologize for the error, the term has been corrected.

  • Table 3 – the “k” in kg-m2 should be lower case

R: It has been corrected.

  • Please add a reference for Eq. 27 – how did you obtain it?

R: Please, see the attached file.

  • Figure 10 is not clear.

R: Figure has been modified by another with higher resolution

Round 2

Reviewer 1 Report

Hi

Thanks for the modifications done.

Reviewer 3 Report

Nil

Minor improvements t required